**communications** engineering

# Actuation of microstructures with spin-current volume effect
Yi-Te Huang[1], Kenta Suzuki[1], Hiroki Arisawa [2], Takashi Kikkawa [2], Eiji Saitoh [2,3,4,5] &
Takahito Ono [1,6] ✉

Microactuators are essential for advances in micro-optics, ultrasonic transducers and microsensors, and there is a growing demand for miniaturization and improved power. Here we demonstrate the actuation of micromechanical structures based on spin-current volume effect using an amorphous magnetic film of TbFeCo with volume magnetostriction. A 2 mm-diameter circular polyimide diaphragm coated with thin TbFeCo/non-magnetic metal films is prepared as the micromechanical structure. When an alternating charge current flows through the TbFeCo/non-magnetic metal films on the diaphragm under an external magnetic field orthogonal to the charge current, an alternating spin-current flows in the non-magnetic metal film due to the spin-Hall effect. In the spin-current volume effect, the spin-current transports angular momentum from the non-magnetic metal to TbFeCo film, and the spin-transfer torque modulates the magnetization fluctuation of the TbFeCo film, causing the diaphragm to vibrate due to spin-lattice coupling. The power density of the TbFeCo/Pt films actuator is larger than $1.17 \times 10^6$ W m$^{-3}$ at 20 mA charge current under 7.2 kOe magnetic field. This value is much higher than that of various existing film-type microactuators. This spin-current volume effect is effective as a new actuation mechanism for microactuators used in micro-optical systems, acoustic diagnostic equipment, and micro-fluidic systems etc.

Recently, microactuators[1–4] have been used in various microdevices, microsensors, and mobile equipment. On the other hand, small actuators have some issues, such as insufficient displacements and powers[1,2,5]. Therefore, research on microactuators with high power and large displacement is underway. Among them, piezoelectric actuators[6–8] are widely used in microdevices, but they require high voltages to drive, have complex fabrication and integration processes, and suffer from long-term durability issues. Magnetostrictive actuators[9–12] have the same or better performance as piezoelectric actuators, including the ability to generate relatively large strain and power even in small sizes. However, magnetostrictive actuators have a couple of disadvantages. Firstly, they rely on electromagnets for control, which are challenging to miniaturize. Secondly, they require electric power to generate and control their magnetic fields.

Recent progress in spintronics has led to the detection of various spin-mechanical interactions, but most of them are related to feeble forces[13–18]. Recently, there have been reports of magnetic material driven by spin-currents[19]. This is a new phenomenon (spin-current volume effect: SVE) in

which a spin-current flows into a magnetic material (TbDyFe) and changes its volume. In this SVE, the volume of the magnetic film changes when a spin-current flows into it. Since this method requires only a permanent magnet and in principle does not require an electromagnet, it is expected to be developed into new microactuators. There are two types of magnetostriction, Joule magnetostriction and volume magnetostriction. In Joule magnetostriction, the rotation of distorted magnetic domains due to the magnetic field is the leading cause of magnetostriction. In volume magnetostriction, spin-lattice coupling is responsible for its magneto-elastic distortion, and the volume magnetostriction is thought to be related to the SVE.

The amorphous TbFeCo (TFC) films exhibit excellent magnetostriction performance (magnetostriction coefficient > 1000 ppm)[20], and it is also reported that electrodeposited FeCo films with Tb impurity indicate noticeable volume magnetostriction[21]. On the other hand, when the amorphous TFC film has a composition of $Tb_{20}Fe_{24}Co_{56}$, Joule magnetostriction becomes small, and the volume magnetostriction becomes the dominant mechanism in magnetostriction (Volume magnetostriction ~ $5.6 \times 10^{-5}$ T$^{-1}$)[10]. In this

[1]Department of Mechanical Systems Engineering, Tohoku University, Sendai 980-8579, Japan. [2]Department of Applied Physics, The University of Tokyo, Tokyo 113-8656, Japan. [3]WPI Advanced Institute for Materials Research, Tohoku University, Sendai 980-8577, Japan. [4]Institute for AI and Beyond, The University of Tokyo, Tokyo 113-8656, Japan. [5]Advanced Science Research Center, Japan Atomic Energy Agency, Tokai 319-1195, Japan. [6]Micro System Integration Center (μSIC), Tohoku University, Sendai 980-8579, Japan. ✉e-mail: takahito.ono.d4@tohoku.ac.jp

study, we fabricated a diaphragm actuator using this amorphous $Tb_{20}Fe_{24}Co_{56}$ as a magnetic thin film with volume magnetostriction, confirmed its actuation by SVE, and evaluated its performance as an actuator.

## Results and discussions

Generally, volume magnetostriction (VM) materials expand (shrink) when it is subjected to a magnetic field for positive (negative) VM materials. Furthermore, as the temperature of a VM material changes, its volume changes because spin-fluctuations and spontaneous magnetization also change. As shown in Fig. 1a, the injection of a spin-current, i.e., a flow of spin angular momentum, into magnetic materials using the spin-Hall effect changes the spin fluctuation due to the spin-transfer torque, causing the volume change in VM materials vian SVE. Non-magnetic metals (NM) such as Pt and W, which have large spin-orbit interactions, are used to generate spin-currents using the spin-Hall effect[22]. The spins accumulate at the interface between the NM and the magnetic films, and the accumulated spins diffuse into the magnetic film, transporting angular momentum.

In this study, diaphragm micromechanical structures are fabricated to evaluate the SVE effect and actuation performance of the TFC film, as shown in Fig. 1b. The device has a 2 mm-diameter circular diaphragm structure consisting of three layers of TFC/NM/polyimide (107 nm/ 100 nm/25 μm in thickness, respectively) with rectangular support. The TFC and NM films are deposited by magnetron sputtering (See Methods). By applying an alternating voltage between the two ends of the TFC/NM layers on the diaphragm sample, a charge current $j_c$ can flow in the TFC/NM layers. The charge current $j_c^{NM}$ flowing in the NM layer contributes to spin-current generation based on the spin Hall effect. The spin-current $j_s$ with spin polarization $\sigma \propto j_c^{NM} \times n$ is injected into the TFC film, where $n$ is a normal vector to the interfacial plane. The direction and amplitude of the alternating spin-current depend on the spin-Hall angle of the NM layer. In experiments, a magnetic field is applied parallel to the diaphragm surface and perpendicular to the alternating charge current. An alternating spin-current flows in the NM layer, and the spin-transfer torque is caused in the

TFC layer. As a result, the magnetization fluctuation changes, and the TFC layer is actuated by SVE. In addition to the SVE force, the Lorenz force $\mathbf{F}_{Lorentz}$ is generated simultaneously, as given by $\mathbf{F}_{Lorentz} = \mathbf{B} \times \mathbf{j}_c L$, where $L$ is the effective current-pass length of the diaphragm. $\mathbf{F}_{Lorentz}$ can be measured from the vibration amplitude using a similar diaphragm sample without the TFC layer by applying an alternating charge current orthogonal to the external magnetic field, and compared with SVE force using the TFC/ NM diaphragm sample.

There are two types of magnetostriction, Joule magnetostriction and volume magnetostriction, and the effect of the spin injection is more pronounced for volume magnetostriction. The volume magnetostriction of the TFC film used in this study is $\sim 5.6 \times 10^{-5}\,T^{-1}$, and the sample shows small Joule magnetostriction in comparison to the volume magnetostriction[10]. In the experiments, the diaphragm is vibrated at its fundamental resonant frequency, and the out-of-plane vibration amplitude is measured at the center of the diaphragm using a Laser Doppler vibrometer (LDV)[19] (See Vibration measurements)

To investigate the NM material dependence on the actuation performance, the driving experiments are performed on four kinds of the diaphragms with four different metal layers on the polyimide film: Pt, TFC/Pt, TFC/Cu, and TFC/W. The Pt diaphragm will be actuated by only the Lorentz force. The spin-Hall angles of Pt and W are reported to be positive and negative values, respectively[22-24], while Cu exhibits a minute spin-Hall angle[22].

For investigating the actuation performance with SVE using magnetostriction film, 107 nm-thick amorphous TbFeCo film is chosen. Figure 2 shows the mechanical response spectra of the four diaphragms driven at 20 mA alternating charge current under a 7200 Oe magnetic field orthogonal to the charge current. The frequency of the alternating charge current is swept, and the peak-to-peak vibration amplitude $A_{PP}$ of the diaphragm is measured by the laser Doppler vibrometer. Each peak with a maximum amplitude represents the fundamental resonance mode of the diaphragm. Figures 3a–d show the amplitude $A_{PP}$ and phase of the diaphragm with the TFC/Pt, TFC/W, TFC/Cu, and Pt layers against frequency in various magnetic fields at a charge current of 20 mA, respectively. The amplitudes in the same phase as the Pt diaphragm driven by the Lorentz force are indicated by negative values, while those of the diaphragm vibrating in the opposite phase are indicated by positive values. In the TFC/Pt diaphragm shown in Fig. 3a, the spin-Hall effect in Pt causes SVE. In Fig. 3c, the TFC/W diaphragm indicates the smaller $A_{PP}$, while its vibration at the resonance is in the opposite phase with those of TFC/Pt and TFC/Cu diaphragms. In Fig. 3d, the Pt diaphragm shows a small $A_{PP}$ than that of the other samples which is caused by only Lorentz force, and the phase is in phase with the vibration of the TFC/W diaphragm.

Figures 4a–d show the charge current $j_c$ dependence on the vibration amplitude $A_{PP}$ under various magnetic fields at the resonance for TFC/Pt,

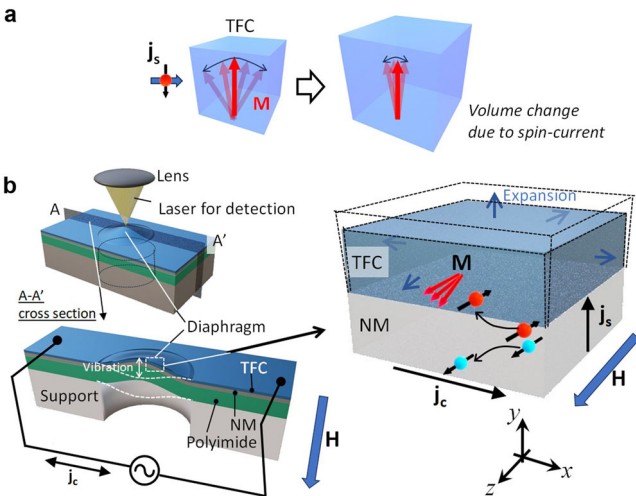

**Fig. 1 | Working principle and experimental method using the diaphragm actuator structures for evaluating the actuation performance of spin-current volume effect (SVE). a** A positive volumetric magnetostriction TbFeCo (TFC) film expands due to the reduction of spin fluctuations caused by the spin transfer torque under an external magnetic field. **b** A schematic illustration of experimental setup using the TFC/NM/Polyimide diaphragm actuated by SVE. **H**, **M**, **j**$_c$, and **j**$_s$ denote the magnetic field, the magnetization of the TFC film, the charge current, and the spin-current. The volume of the TFC film can be modulated by the spin transfer torque via spin-current injection. Here, an electron with its spin (magnetic moment) antiparallel (parallel) to the magnetization **M** is injected into the TFC film. Thus, an alternating charge current flow in the TFC/NM films induces mechanical vibration in the diaphragm. The right-side figure shows the schematic when Pt is used as the NM layer.

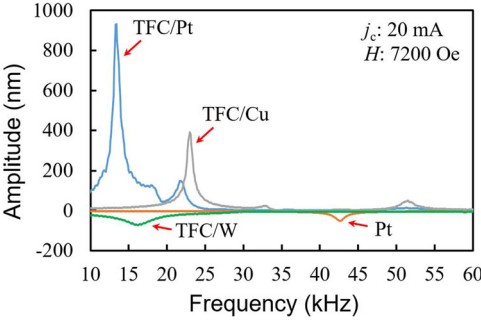

**Fig. 2 | Vibration amplitudes (peak-to-peak value) measured by sweeping the frequency of an alternating charge current applied to the diaphragm sample (metal layers: TFC/Pt, TFC/W, TFC/Cu, Pt) and applying an external magnetic field orthogonal to the alternating charge current, where TFC is TbFeCo.** The measurements are performed with alternating charge current $j_c$ = 20 mA at magnetic field $H$ = 7200 Oe. The main vibration peak corresponds to $f_{01}$ mode, and the second peak corresponds to $f_{11}$ mode.

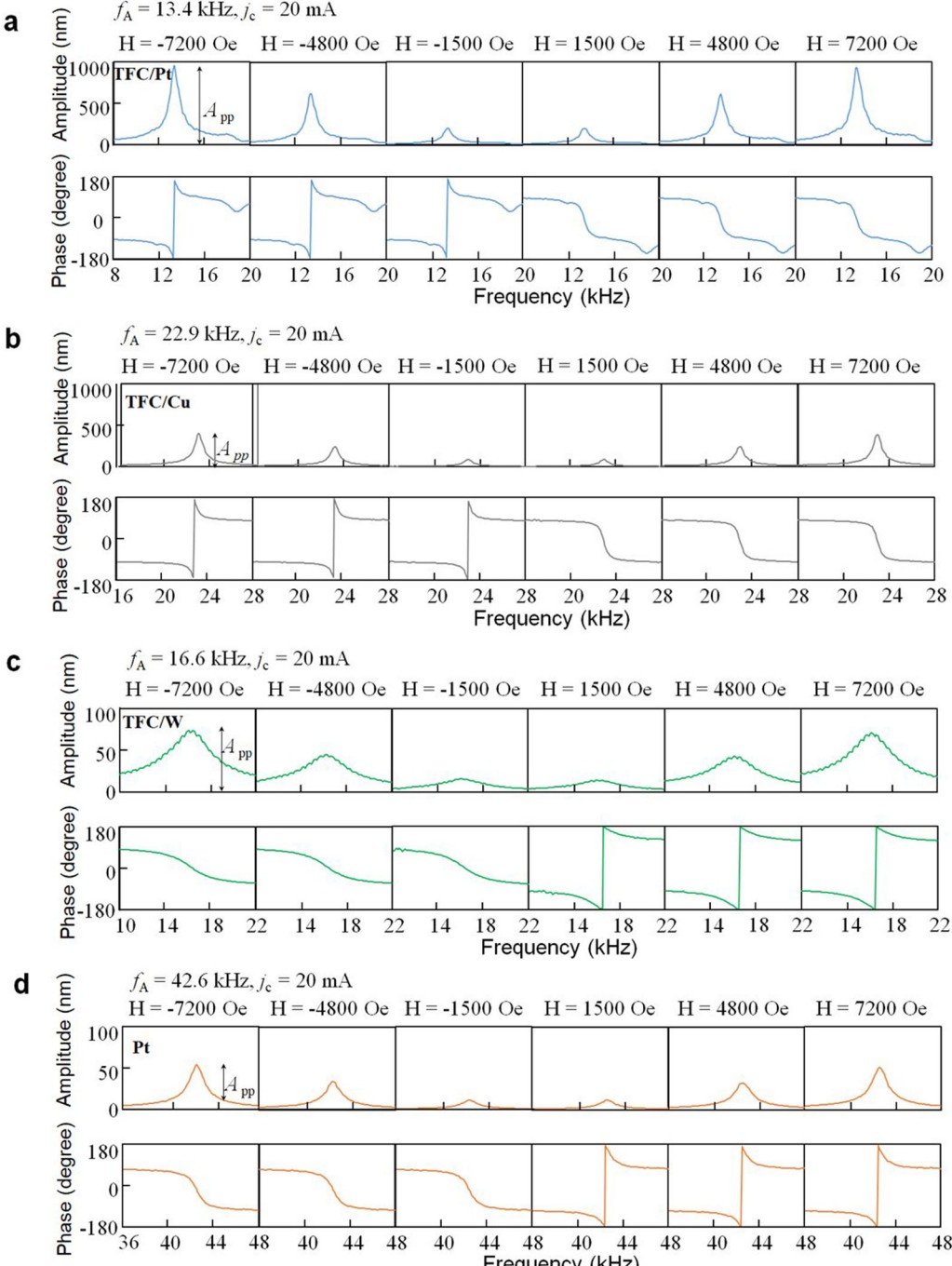

**Fig. 3 | Typical mechanical response spectra of the amplitude and phase for the diaphragm actuators. a** TFC/Pt, **b** TFC/Cu **c** TFC/W, and **d** Pt diaphragm at various magnetic fields (**H**), where TFC is TbFeCo. The measurements are performed with alternating charge current $j_c$ = 20 mA. The frequency $f_A$ at maximum vibration amplitude corresponds to the fundamental resonant frequency.

TFC/Cu, TFC/W, Pt diaphragms, respectively. All samples show a linear relationship between $A_{PP}$ and $j_c$. The charge current flowing in the NM layer causes the spin Hall effect, which can be calculated from each resistivity and thickness (See Methods). If considering the amplitude per charge current in the NM layer, the TFC/W sample exhibits a comparable value to that of the TFC/Pt sample, but its vibration phase is opposite, as shown in Fig. 1S in Supplementary Information.

The magnetic field and current are parallel, the magnetization fluctuation is not modulated, and no apparent vibrations due to SVE are observed, as shown in Fig. 5[19]. Those results show that the phase of the vibration depends on the direction of the magnetic field and NM materials.

In the TFC/Pt sample with a positive spin-Hall angle of Pt, the spin-current causes the volumetric expansion in the TFC film with positive volume magnetostriction due to SVE. Since the TFC/Pt layer is on the thick polyimide layer, the expansion of the TFC layer causes the diaphragm to deflect with the convex side upward. In this case, the direction of deflection of the diaphragm is opposite to the direction of Lorentz force. On the other hand, in the TFC/W sample with a negative spin-Hall angle of W, the phase of the spin-current induced volume expansion is opposite to that of the TFC/Pt sample. Therefore, the TFC/W diaphragm vibrates in the opposite phase with the TFC/Pt diaphragm. In all samples, reversing the direction of the external magnetic field inverts the phase of the vibrations, and the vibration

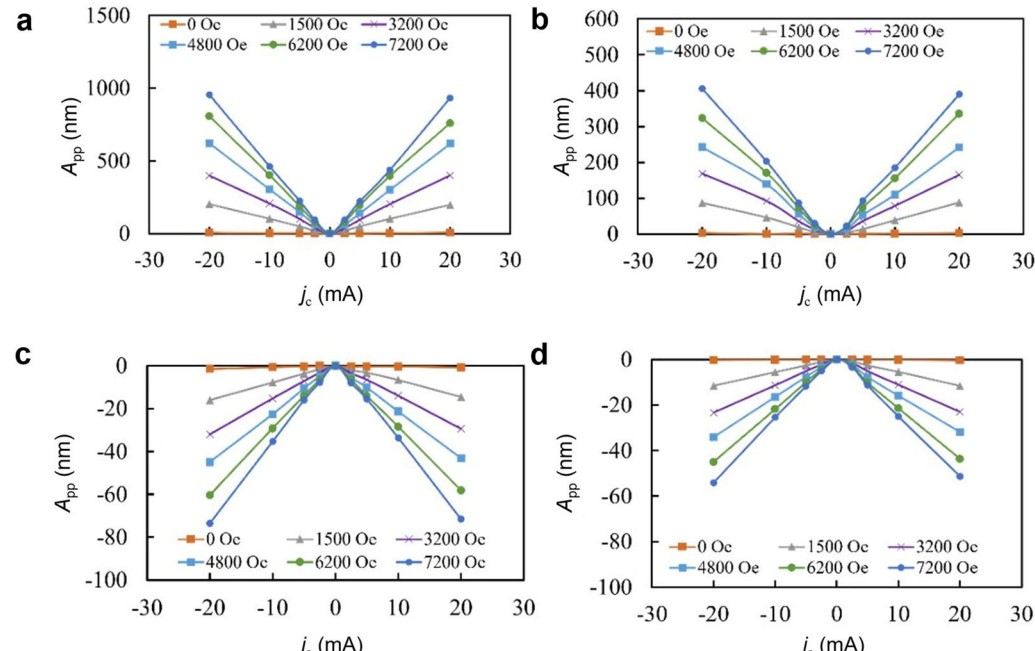

**Fig. 4 | Experimental results on the dependence of actuation on charge currents.** Relationship between the vibration amplitude $A_{pp}$ and the charge current $j_c$ at various magnetic fields for **a** TFC/Pt, **b** TFC/Cu, **c** TFC/W, and **d** Pt diaphragm, where TFC is TbFeCo. The amplitudes $A_{pp}$ in the same phase as the Pt diaphragm driven by the Lorentz force are indicated by negative values, while those of the diaphragm vibrating in the opposite phase are indicated by positive values.

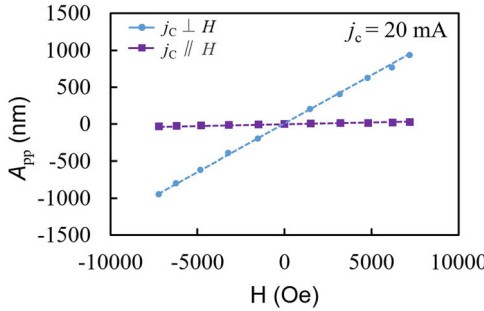

**Fig. 5 | Experimental result on the dependence of actuation on magnetic field direction.** Magnetic field direction dependence on the vibration amplitude of the resonance for the TbFeCo/Pt diaphragm. The peak-to-peak vibration amplitudes ($A_{pp}$) for cases of magnetic fields ($H$) parallel and perpendicular to the alternating charge current ($j_c$) are compared.

amplitude is proportional to the charge current, as expected from SVE. Those results suggest that the TFC films exhibit the SVE. If the TFC film is under the NM film, the phase is also considered to be inverted.

The vibration amplitude of the resonance is amplified by the Q factor in the resonance because the generated force energy is stored as mechanical vibration energy. At the resonance peak frequency, the generated force $F$ is given by

$$F = kA/Q, \tag{1}$$

where $k$ is the spring constant of the diaphragm, $A$ is the vibration amplitude at the center of the diaphragm, and $Q$ is the Q factor[25]. The spring constant $k$ of the circular diaphragm with tension is given by[26]

$$k = \frac{16\pi E t^3}{3R^2(1-\nu^2)} + 4\pi\sigma t, \tag{2}$$

where $E$ is the Young's modulus of the composite diaphragm, $t$ is the thickness of the diaphragm, $R$ is the radius of the diaphragm, $\nu$ is the Poisson's ratio of the diaphragm, $\sigma$ is the residual stress of the diaphragm. Effective Young's moduli are used as Young's moduli of the composites diaphragm (Supplementary Note 2). The Pt film has residual tensile stresses ~$50 \pm 20$ MPa, and the effect of residual stresses cannot be ignored in the estimation of spring constants. If stresses are taken into account in the calculation only in the case of Pt diaphragm, the spring constants of the diaphragm TFC/Pt, TFC/Cu, TFC/W, Pt are calculated to be $8.46 \times 10^2$, $7.67 \times 10^2$, $1.12 \times 10^3$, $1.65 \times 10^4 \pm 0.6 \times 10^4$ N m$^{-1}$ using Eq. (2) (Supplementary Note 3).

In order to compare the generated force due to SVE and Lorentz force, it is more convenient to normalize the vibration amplitude by the Q factor. The Q factor of each resonance peak is obtained from the ratio of peak frequency $f_0$ to $-3$ dB-width $\Delta f$ of maximum amplitude, $Q = f_0/\Delta f$. The Q factor depends on the materials because of their internal loss, ranging from 5 to 38 (Supplementary Note 3).

Figure 6a shows the magnetic field dependence on $A_{pp}$, and Fig. 6b shows the $A_{pp}$ divided by the Q factor, $A_{pp}/Q$, at 20 mA charge current. The generated force by SVE is expected to be proportional to the projection of the magnetization **M** along **H** // **z** [in Fig. 1a] and the flowing charge current. This TFC film has an easy axis of magnetization in the out-of-plane direction due to the stress field, and the in-plane magnetization is proportional to the magnetic field within the measured range[27]. In general, the relationship between force $F$ and tensile volumetric strain $\varepsilon$ in a diaphragm constrained around its perimeter is given by $F = -2c\varepsilon A$, where $c$ is the modulus of volume elasticity of the magnetic material, $A$ is the area of the diaphragm. The generated strain $\varepsilon$ is isotropic in unconstrained uniform magnetic materials, and the force is generated in the direction of constraint in the plane of the magnetic film diaphragm. On the other hand, in the experiment, large vibrations are obtained despite the small value of the spin-Hall angle of Cu. It may originate in CuO$_x$ at the TFC-Cu interface, which has a large spin Hall angle[28]. A portion of the total charge current flowing in the NM layer contributes to the generation of the spin-current by the spin Hall effect. The W layer has ~40 times larger resistivity than that of the Pt layer, possibly due to the oxygen impurity during sputtering; thus, the strain generated by SVE becomes a small value due to the small charge currents ~1.65 mA in the W layer. The vibration amplitudes normalized by the charge current for the TFC/W diaphragm show

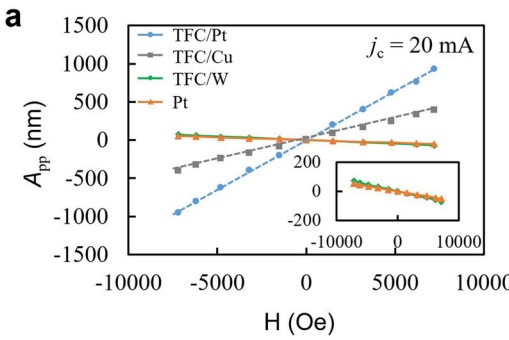
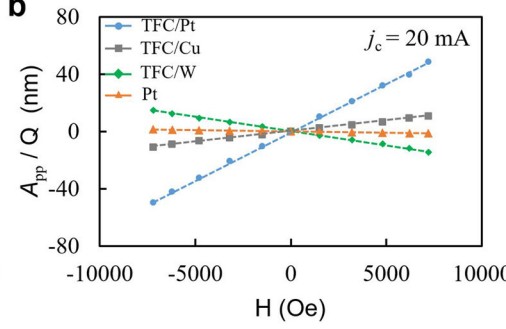

**Fig. 6 | The vibration amplitudes ($A_{pp}$) are normalized by the quality factor (Q) of mechanical resonance, which is proportional to actuation force. a** Magnetic field (H) dependences of $A_{pp}$ at the resonance for all samples at a charge current $j_c$ = 20 mA. The inset is the magnified graph plotted on only TFC/W and Pt diaphragms, where TFC is TbFeCo. **b** Magnetic field dependence of $A_{pp}$ normalized by Q factor, $A_{pp}/Q$, at a charge current $j_c$ = 20 mA.

comparable values with that of the TFC/Pt diaphragm (See Supplementary Note, Fig. S1).

Using Eq. (1) (2), generated forces due to SVE and Lorentz force can be calculated (Supplementary Note 3). The actual force observed should be the vector sum of the forces due to SVE and the Lorentz forces. In other words, if the vibration is in the opposite phase to the vibration due to the Lorentz force, the actual force generated by SVE is the sum of the observed force and the Lorentz force. Thus, in the case of TFC/Pt diaphragm, the generated force is estimated to be 63.4 μN at 20 mA charge current under 7200 Oe magnetic field. This generated force is 2.9 times larger than that of Lorentz force (~−22 μN) observed in the Pt layer. The power density $P_i$ of the actuator can be calculated as follows,

$$P_i = \frac{FA_{pp}f}{2V},\qquad(3)$$

where $A_{pp}$ is the peak-to-peak vibration amplitude, $f$ is the driving frequency, $V$ is the volume of the actuation material. If it is supposed that only the TFC film produces the actuation power, the power density is calculated to be $1.17 \times 10^6$ W m$^{-3}$ in the TFC/Pt diaphragm. Generally, the spin-current diffusion length is of an order of 10 nm, thus the strain due to SVE possibly happening in the TFC layer near the NM/TFC interface. If the distortion is caused by SVE in the 10 nm-depth from the interface in the TFC, its power density is estimated to be $1.24 \times 10^7$ W m$^{-3}$. In general, power density depends on the actuator configuration, and it is difficult to obtain a large power density using thin film actuators[29]. However, the power density produced by SVE can be compared with that of actuators with a similar structure. For example, actuators with 200 μm-thick PZT membranes have been reported to produce a power density of 10,000 W m$^{-3}$[30]. A stacked-PZT membrane actuator has been shown to produce ~800 W m$^{-3}$ of power density[31]. Compared to other thin film actuators, this TFC film driven by the SVE can produce a very high power density even with the diaphragm with a thickness of 107 nm.

## Conclusions

In conclusion, we found that a sputtered amorphous metal (TFC: TbFeCo) with volume magnetostriction exhibits the spin-current volume effect (SVE). The actuation characteristics based on the SVE were evaluated from the vibration displacement of a bi-material structure of a non-magnetic (NM) film and the TFC film on a polyimide diaphragm with a constrained periphery. When a charge current is applied to the metal on the diaphragm in a magnetic field perpendicular to the charge current, a spin-current is generated in the non-magnetic material with a large spin-orbit interaction, and as a result, the spin-transfer torque causes a change in magnetization in the volume magnetostriction material, which is coupled with magnetoelastic effect. The spin-transfer torque changes the magnetization fluctuation in the magnetic material, which generates strain due to magnetoelastic coupling.

This SVE actuator was found to have a large power density despite using the very thin magnetic film. This SVE actuator is expected to be very useful for microactuators and microsensors with electromechanical transduction.

## Methods

### Sample preparation

A magnetron sputtering process is employed to deposit the TFC and non-magnetic layers on a 25 μm-thick polyimide plate. After the deposition of the NM layer on the polyimide substrate, the TFC film is sputtered at 5 Pa of working Ar gas pressure with an RF power of 50 W for 20 min at room temperature on the polyimide plate while the base pressure is lower than $10^{-4}$ Pa. The TFC film sputter-deposited at 5 Pa exhibits volume magnetostriction of ~43 ppm at 7490 Oe[27]. After the deposition of the TFC films, Energy Dispersive X-ray Analysis (EDX) is employed to analyze the elemental composition and thickness of sputtered TFC films[21]. For fabricating a diaphragm structure, the polyimide plate coated with the TFC/NM film is cut into 8 mm × 4 mm and bonded with an aluminum plate with a 2 mm-diameter hollow using an adhesive. As an NM film, a 100 nm-thick Pt, Cu, or W film with 5 ~ 20 nm-thick adhesive titanium layer is deposited on the polyimide substrate by magnetron sputtering before depositing the TFC film. The resistivity of each film is measured by a four-terminal method. The resistivities of TFC, Pt, Cu, W, and Ti layers are $2.47 \times 10^{-6}$, $4.45 \times 10^{-7}$, $3.36 \times 10^{-8}$, $1.76 \times 10^{-5}$ and $1.0 \times 10^{-6}$ Ωm, respectively. The resistivity of the W layer is much bigger than that of the bulk value (~$5.6 \times 10^{-8}$ Ωm), which may be caused by slight oxidization during sputtering. The actual charge current flowing through the NM layer is determined by the resistivities and thicknesses of each layer comprising the diaphragm.

Electrical wires connecting to the sample are glued on a rigid sample stage and embedded into an epoxy to prevent the influence of the wire vibration.

### Vibration measurements

Mechanical vibrations of the diaphragm are measured at the center of the diaphragm by using a laser Doppler vibrometer (Custom-made Double laser Doppler system from Neoark Co., Japan) via an objective lens (x10). The mechanical vibration signal is detected using the lock-in method (EG&G 7260). For the measurement of the spin-current volume effect, the voltage from a signal generator (NF corporation, WF1948) is applied to samples via a laboratory-made current amplifier. An external magnetic field is applied using an electromagnetic coil (GMW, model 3480) equipped with the laser Doppler vibrometer. All measurements are performed at room temperature and ambient pressure. In the lock-in method, no second harmonic signals are observed in the mechanical vibration, indicating no electrothermal influence on the induced vibration. In order to ensure no thermal effect, the charge current applied to the sample is limited to 20 mA.

**Article**

## Data availability

The data that support the findings of this study are available from the corresponding author upon reasonable request.

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

## Acknowledgements

Part of this research was performed at Tohoku University Micro System Integration Center and Micro / Nano-Machining Research and Education Center. We would like to acknowledge to Mr. N. Tamazawa for his support. This work is partly supported by JST, the establishment of university fellowships towards the creation of science technology innovation, Grant Number JPMJFS2102, JST-CREST (JPMJCR20C1 and JPMJCR20T2), Grant-in-Aid for Scientific Research (JP19H05600, JP20H02599, and JP22K18686) and Grant-in-Aid for Transformative Research Areas (JP22H05114) from JSPS KAKENHI, MEXT Initiative to Establish Next-generation Novel Integrated Circuits Centers (X-NICS) (JPJ011438), Japan, and Institute for AI and Beyond of the University of Tokyo.

## Author contributions

Y.H. carried out the experiments and analyzed the data with the help of T.O., K.S. prepared the experimental setup, Y.H. and T.O. prepared the manuscript with the help of H.A., T.K., and E.S. All the authors discussed the results and commented on the manuscript.

## Competing interests

The authors declare no competing interests.
