## [Peer Review File · Communications Engineering]

Reviewers' comments:

Reviewer #1 (Remarks to the Author):

In this paper, the authors study the actuation of micromechanical structures based on spin-current volume effect (SVE) using an amorphous TbFeCo film.

The SVE has been studied by the same group by using TbDyFe film (Ref. 19). The present work is an extension of Ref. 19 by using TbFeCo.

The paper is interesting and well written.

But, there are the following questions:

1) Why is an amorphous TbFeCo film used instead of FeDy Fe in this work?

2) NM is defined to be non-magnetic metal.

But in the Results and discussion, it is noted to be non-metallic film.

This must be a typesetting error.

When the above questions are answered, the revised manuscript may be published in Communications Engineering.

Reviewer #2 (Remarks to the Author):

In the previous paper (Nature Commun. 13, 2440 (2022)), the authors showed that the spin current generated by the spin Hall effect can induce a volume magnetostriction, which they named the "spin-current volume effect (SVE)". In this paper, they have investigated the performance of mechanical actuators driven by SVE in suspended TbFeCo/paramagnetic metal bilayer films formed on polyimide diaphragms. This paper is a pioneering work that demonstrates potential applications of spin mechanics. Therefore, I recommend the publication of this paper in Commun. Eng. if the following issues are addressed.

1) In addition to SVE driven by the spin Hall effect and the Lorentz force, the contribution of the spin Seebeck effect should also be discussed. Since the electrical conductivity of the TbFeCo films and the paramagnetic metals are significantly different, the in-plane charge current can induce a vertical temperature gradient, which may drive spin currents and SVE.

2) The effect of the sample heating should be discussed in more detail. In the suspended structures on polyimide diaphragms, the sample temperature should be much higher than the ambient temperature due to Joule heating because of the absence of heat transfer to substrates. Since the electrical conductivities of Pt, W, and Cu are different by orders of magnitude, the Joule heating contribution and the sample temperature should be quite different. If the temperatures differ significantly, the effect of thermal expansion and the magnitude of magnetization of the TbFeCo layers may also change.

3) The thicknesses of the TbFeCo layers (107 nm) and the paramagnetic metal layers (100 nm) are much larger than their spin diffusion lengths, except for Cu. As stated on page 8, SVE should occur near the TbFeCo/paramagnetic metal interfaces if it is driven by spin currents. To confirm this interpretation, it should be checked that the actuation performance does not change even if the TbFeCo layer is thinner.

4) In the experiments reported, the charge current applied to the bilayers is fixed at 20 mA. However, SVE driven by the spin Hall effect is proportional to the current density in the paramagnetic metal layers. The authors should estimate the current density in the paramagnetic metal layers by assuming a simple equivalent circuit, normalize the vibration amplitude by the obtained values, and compare the results for different samples. Since the authors have already measured the electrical conductivity of each layer, this can be easily performed.

5) On page 7, the authors state "It is expected that the generated force by SVE is proportional to the magnetization". Does this mean that SVE is proportional to the saturation magnetization? If so, the origin of the magnetization dependence should be discussed. If this statement merely describes the symmetry of Eq. (3), the sentence should be corrected as it may mislead readers.

6) The data in Supplementary Information are very important. At least, Figs. S2-S5 should be moved to the main text. Figures S3 and S4 can be combined with Figs. 3 and 4, respectively.

7) Why were the TbFeCo films deposited at 5 Pa? This pressure is more than an order of magnitude higher than typical pressure for magnetron sputtering.

Reviewer #3 (Remarks to the Author):

In the manuscript by Y.-T. Huang et al., the authors report the spin-current volume effect (SVE) in the amorphous magnetic film of Tb₂₀Fe₂₄Co₅₆ (TFC) with spin current generated from the spin Hall effect in heavy elements such as Pt and W. This work is similar to their previous work [Nat. Comm. 13, 2440 (2022)] but with a different magnetic material. They observe opposite expansion or shrinking results for TFC/Pt and TFC/W samples indicating a spin current origin. The methods are solid and the study of the influence of spin current on the mechanical property of a material is pioneering. Therefore, I could recommend the publication of this work in Comm. Eng. if the authors could address my following questions.

1. Fig. 1 is counter-intuitive. An opposite angular momentum in the -z direction, that delivers into the magnetic film, which is magnetized in the +z direction, reduces the spin fluctuation. I would think it makes the fluctuation even more severe until it switches the magnetization. Can the authors comment on this? Is it possible that a "-" sign is missing somewhere?

2. Different from their previous work on Tb_{0.3}Dy_{0.7}Fe₂ (TDF) diaphragm, the four diaphragm samples with or without TFC shown in Fig. 2 in this work each have a different resonance frequency. Is there a reason?

3. In the case of TFC/Pt and TFC/Cu, it looks like there is more than one resonance peak. Can the authors comment on this result?

4. Why the SVE is significant in TFC/Cu, but not in the TDF/Cu case?

5. Can the authors compare the SVE efficiency on the actuation amplitude between TFC in this work and TDF in the previous work?

6. The authors mention at the bottom of page 8, that "it is difficult to obtain a large power density in thin films", but their Eq. 5 shows that power is inversely proportional to thickness, which is contradictory to

their conclusion. Can the authors explain in detail their claim?

7. In the “Methods” session, the resistivity of the four films shall have a physical unit. I guess the unit is Ωm . In this case, the resistivity of $1760\ \mu\Omega\text{cm}$ for the 100-nm thick W film is too high compared to other reports. W could be easily oxidized. Can the authors explain why they have such a high resistivity for W film?

We deeply appreciate the reviewers for providing kind and meaningful questions and suggestions on the manuscript. We believe that the manuscript has improved very much according to the comments. Revised parts are indicated by color fill.

Reviewer #1 (Remarks to the Author):

In this paper, the authors study the actuation of micromechanical structures based on spin-current volume effect (SVE) using an amorphous TbFeCo film. The SVE has been studied by the same group by using TbDyFe film (Ref. 19). The present work is an extension of Ref. 19 by using TbFeCo. The paper is interesting and well written. But, there are the following questions:

1) (comments from the reviewer) *Why is an amorphous TbFeCo film used instead of FeDy Fe in this work?*

(Reply) Thank you for your comments. We are still working on research of the actuation performance of SVE of TbDyFe thin films. It takes time to prepare the same quality of the TbDyFe films, the exact report will appear in the future.

2) *NM is defined to be non-magnetic metal. But in the Results and discussion, it is noted to be non-metallic film. This must be a typesetting error. When the above questions are answered, the revised manuscript may be published in Communications Engineering.*

(Reply) Thank you for your kind pointing. We revised the paper according to the comment, “Metallic” is revised to “Magnetic”.

Reviewer #2 (Remarks to the Author):

In the previous paper (Nature Commun. 13, 2440 (2022)), the authors showed that the spin current generated by the spin Hall effect can induce a volume magnetostriction, which they named the "spin-current volume effect (SVE)". In this paper, they have investigated the performance of mechanical actuators driven by SVE in suspended TbFeCo/paramagnetic metal bilayer films formed on polyimide diaphragms. This paper is a pioneering work that demonstrates potential applications of spin mechanics. Therefore, I recommend the publication of this paper in Commun. Eng. if the following issues are addressed.

1) (Reviewer comment) *In addition to SVE driven by the spin Hall effect and the Lorentz force, the contribution of the spin Seebeck effect should also be discussed. Since the electrical conductivity of the TbFeCo films and the paramagnetic metals are significantly different, the in-plane charge current can induce a vertical temperature gradient, which may drive spin currents and SVE.*

(Reply)

Thank you for your important suggestion.

Spin currents due to the spin Seebeck effect appear in 2ω , but we observe 1ω . Therefore, the spin Seebeck effect is not superimposed on the signal. As the reviewer suggested, Joule heating is considered for the bi-metal layer, and the temperature gradient is generated in the out-of-plane direction. In experiments, we evaluated 2nd harmonic component of the mechanical vibration using the lock-in amplifier, but no meaningful signal was observed. Therefore, we are considering that

thermal influence can be negligible. In the paper, we added the following sentence.

(12 page, 12 line)

In the lock-in method, no second harmonic signals are observed in the mechanical vibration, indicating no electrothermal influence on the induced vibration.

2) The effect of the sample heating should be discussed in more detail. In the suspended structures on polyimide diaphragms, the sample temperature should be much higher than the ambient temperature due to Joule heating because of the absence of heat transfer to substrates. Since the electrical conductivities of Pt, W, and Cu are different by orders of magnitude, the Joule heating contribution and the sample temperature should be quite different. If the temperatures differ significantly, the effect of thermal expansion and the magnitude of magnetization of the TbFeCo layers may also change.

(Reply)

Thank you for your important comments. We examined, Joule heating influence using a TDF/Pt diaphragm sample (TFC 54 nm, Pt 100 nm), and it is found that the electrothermal influence is observed at over 150 mA of current. In the experiments, a second harmonic of vibration can be detected, and the mechanical frequency shift is observed at 150 mA of current. In our paper, in order to ensure no thermal effect, the current is limited to 20 mA. In addition, no second harmonic of vibration is observed from lock'in detection as above, which is evidence of a less thermal effect.

We add the following sentence.

(12 Page, 14 line) In order to ensure no thermal effect, the charge current applied to the sample is limited to 20 mA.

3) The thicknesses of the TbFeCo layers (107 nm) and the paramagnetic metal layers (100 nm) are much larger than their spin diffusion lengths, except for Cu. As stated on page 8, SVE should occur near the TbFeCo/paramagnetic metal interfaces if it is driven by spin currents. To confirm this interpretation, it should be checked that the actuation performance does not change even if the TbFeCo layer is thinner.

(Reply) The thickness dependence of the TFC film is given by the right figure. We think that there is no obvious thickness dependence.

It should be noted that we examined the TFC/Pt sample, but the stress of the TFC/Pt film was compressive (-280 MPa), which is different from the sample shown in the paper (-34 MPa), where the Pt film has different stress. The quality factors of resonance are ~ 20. The actuation power of SVE possibly depends on the stress of the film. The performance of the TFC film is very sensitive to the sputtering condition, as referred our paper [27], which may cause the performance fluctuation of the TFC film.

In more detail, we need to perform more experiments. We are thinking that the stress dependence and exact thickness dependence would be our future work.

4) In the experiments reported, the charge current applied to the bilayers is fixed at 20 mA. However, SVE driven by the spin Hall effect is proportional to the current density in the paramagnetic metal layers. The authors should estimate the current density in the paramagnetic metal layers by assuming a simple equivalent circuit, normalize the vibration amplitude by the obtained values, and compare the results for different samples. Since the authors have already measured the electrical conductivity of each layer, this can be easily performed.

(Reply) Thank you for your important suggestions. When the applied total currents are 20 mA, the current j_{MN} flowing in the Pt film in TFC/Pt layer, Cu film in TFC/Cu, W film in TFC/W are estimated to be 15.6, 19.9, 1.65, respectively. The thicknesses of the NM layers are 100 nm thick; therefore, the current densities becomes also this ratio. It should be noted that the influence of the Ti adhesion layer is also considered. The magnetic field dependence of A_{pp} normalized by the current flowing in NM film at the resonance for all samples are plotted in Fig. 1S in Supplementary note. The current density of the W layer is small because of the high resistivity. It can be seen that the absolute value of A_{pp}/j_{MN} of the TFC/W sample is comparable to that of the TFC/Pt sample. We think that the W film may be oxidized, thus, the resistivity increased. The paper is revised as follows.

1. Amplitude normalized by charge current in the NM film is shown in Fig. 1S (Supplementary note).
2. (10 page, 6 line) The W layer has ~ 40 times larger resistivity than that of the Pt layer, possibly due to the oxygen impurity during sputtering; thus, the strain generated by SVE becomes a small value due to the small charge currents ~1.65 mA in the W layer. The vibration amplitudes normalized by the charge current for the TFC/W diaphragm show comparable values with that of the TFC/Pt diaphragm (See Supplementary note, Fig. S1).
3. (11 page, 29 line) The resistivities of TFC, Pt, Cu, W and Ti layers are 2.47×10^{-6} , 4.45×10^{-7} , 3.36×10^{-8} , 1.76×10^{-5} and 1.0×10^{-6} Ωm , respectively.

5) On page 7, the authors state "It is expected that the generated force by SVE is proportional to the magnetization". Does this mean that SVE is proportional to the saturation magnetization? If so, the origin of the magnetization dependence should be discussed. If this statement merely describes the symmetry of Eq. (3), the sentence should be corrected as it may mislead readers.

(Reply) Thank you for your suggestion. The expression was not correct. As reported in our previous paper, SVE is proportional to magnetization fluctuation ΔM . The displacement given by SVE is already published by our previous paper [19], we removed this equation (3) from the paper. Sentence is changed as follows.

(9 page, 22 line) magnetization -> projection of the magnetization M along $H // z$ [in Fig. 1(a)].

6) The data in Supplementary Information are very important. At least, Figs. S2-S5 should be moved to the main text. Figures S3 and S4 can be combined with Figs. 3 and 4, respectively.

(Reply) Thank you for your important suggestion. According to the reviewer's comments, Figures

S3 and S4 are combined with Figs. 3. Figure 4S is combined with Fig. 6. Figure S2 is moved to Figure 4.

7) Why were the TbFeCo films deposited at 5 Pa? This pressure is more than an order of magnitude higher than typical pressure for magnetron sputtering.

(Reply) Thank you for your important point. The details of the properties of the TbFeCo film for sputtering conditions are described in reference 27. In our sputtering system, the films tend to show strong compressive stress at low Ar pressure ~ 1 Pa. It is found that for the large actuation performance, a low-stress film is essential, but if the sputtering pressure is high, the density becomes low. In addition, a pressure dependence on volume magnetostriction. We find that the film sputtered at 5 Pa shows maximum volume magnetostriction (27). Thus, we chose the sputtering Ar pressure ~ 5 Pa.

We add the following sentence.

(11 page, 21 line) The TFC film sputter-deposited at 5 Pa exhibits volume magnetostriction of ~ 43 ppm at 7490 Oe [27].

Reviewer #3 (Remarks to the Author):

In the manuscript by Y.-T. Huang et al., the authors report the spin-current volume effect (SVE) in the amorphous magnetic film of Tb₂₀Fe₂₄Co₅₆ (TFC) with spin current generated from the spin Hall effect in heavy elements such as Pt and W. This work is similar to their previous work [Nat. Comm. 13, 2440 (2022)] but with a different magnetic material. They observe opposite expansion or shrinking results for TFC/Pt and TFC/W samples indicating a spin current origin. The methods are solid and the study of the influence of spin current on the mechanical property of a material is pioneering. Therefore, I could recommend the publication of this work in Comm. Eng. if the authors could address my following questions.

1) *Fig. 1 is counter-intuitive. An opposite angular momentum in the -z direction, that delivers into the magnetic film, which is magnetized in the +z direction, reduces the spin fluctuation. I would think it makes the fluctuation even more severe until it switches the magnetization. Can the authors comment on this? Is it possible that a “-” sign is missing somewhere?*

(Reply) Thank you for your question. Since the electron has a negative gyromagnetic ratio, the magnetic moments opposite to the angular momentum of electron spin are transferred to the magnetic moments of TFC as an angular momentum change. Therefore, we consider that the spin-current generated by the spin Hall effect enlarges the magnetization, i.e. reduces magnetization fluctuation.

We add the following sentence in figure caption of Fig. 1

(3 page, 9 line) Here, an electron with its spin (magnetic moment) antiparallel (parallel) to the

magnetization M is injected into the TFC film.

2) *Different from their previous work on Tb_{0.3}Dy_{0.7}Fe₂ (TFC) diaphragm, the four diaphragm samples with or without TFC shown in Fig. 2 in this work each have a different resonance frequency. Is there a reason?*

(Reply) Thank you for your essential point. In our previous research of TDF, we didn't use diaphragm structures. The TDF film is formed on a rigid Si wafer. The displacement measurements were performed only on the film thickness change at off-resonance, which is different from this paper.

The resonant frequency of the diaphragm depends on the stress of the film. In addition, the TFC/NM/polyimide films are bonded to the base metal, which also induces bonding stress. The TFC films have compressive stress, and the compressed films are difficult to actuate. Thus, we chose tensile-stressed NM films. By making the bi-layer structure, the stress can be reduced. However, stress difference and Young's moduli difference exist. The diaphragm with Pt only shows tensile stress (50 MPa), which makes the spring of the diaphragm hard, as shown in Eq. (2). As a result, the resonant frequency is increased much. In order to consider this spring hardening effect, the stress effect of the Pt film is considered for power density calculation. The details on stress influence are described in Supplementary Note 3.

3) *In the case of TFC/Pt and TFC/Cu, it looks like there is more than one resonance peak. Can the authors comment on this result?*

(Reply) Thank you for your important comment. It looks like a higher mode of vibration of diaphragm. Main peak corresponds to f_{01} mode, the second mode is f_{11} mode, which is 1.59 times higher than that of f_{01} mode. Third peak will be f_{02} mode, which is 2.3 times higher than f_{01} mode. However, the exact frequency is also related to film stress; therefore, the frequency of each mode is slightly different from the expected values.

In the figure caption of Fig. 2. We add the following sentence.

(5 Page 5 line) "The main vibration peak corresponds to f_{01} mode, and the second peak corresponds to f_{11} mode."

4) *Why the SVE is significant in TFC/Cu, but not in the TDF/Cu case?*

(Reply) Thank you for your question. The sputtering of Cu and TFC films is performed by different sputtering machines. Therefore, the Cu film is exposed to the atmosphere, which possibly causes CuOx. Cu oxidation is reported to enhance the spin-Hall angle (28). In contrast, TDF deposition is performed by electrodeposition using an acid electrolyte. The acid electrolyte may remove the Cu oxide layer at the moment of deposition starting.

We added the following sentence.

(10 Page, 3 line) "It may originate in CuOx at the TFC-Cu interface, which has a large spin Hall angle [28]."

28. H. An, Y. Kageyama, Y. Kanno, N. Enishi, and K. Ando, "Spin-torque engineered by natural

oxidation of Cu”, *Nature Communications*, **7** (13069), 1-8 (2016)

5) Can the authors compare the SVE efficiency on the actuation amplitude between TFC in this work and TDF in the previous work?

(Reply)

We are currently working on the actuation experiment on TDF using similar diaphragm. It is not yet concluded; it will be reported in our future publication.

6) The authors mention at the bottom of page 8, that “it is difficult to obtain a large power density in thin films”, but their Eq. 5 shows that power is inversely proportional to thickness, which is contradictory to their conclusion. Can the authors explain in detail their claim?

(Reply)

Thank you for your important pointing. Equation 5 did not include shape effects, which made the subsequent explanation difficult to understand.

For example, a twice thicker actuator can produce twice force and twice displacement in thickness direction. Thus, 4-times bigger power can be produced, However, there is a material limitation, this can be adapted for thin actuator. It has been reported that a stacked-type PZT actuator can provide a power density two orders of magnitude higher than that of bender-type thin-actuators (29), as shown figure below. Stacked actuator allows the stresses in each layer to add up during actuation, generating greater displacements and forces, thus increasing the energy density compared to single-layered actuators.

We modified the sentence and added the above reference in the manuscript as follows.

(10 Page, 28 line) In general, power density depends on the actuator configuration, and it is difficult to obtain a large power density using thin film actuators [29].

[29] Jose L. Pons, “Emerging actuator technologies: a micromechatronic approach”, John Wiley & Sons, 2005.

Figure 2.32 Relative position of piezoelectric stacks, benders, inchworm actuators and TWUMs on the energy–power density plane.

7) In the “Methods” session, the resistivity of the four films shall have a physical unit. I guess the unit is Ωm . In this case, the resistivity of 1760 $\text{m}\Omega\text{cm}$ for the 100-nm thick W film is too high compared to other reports. W could be easily oxidized. Can the authors explain why they have such a high resistivity for W film?

(Reply) Thank for your important pointing. We add the unit “ Ωm ”. Maybe our machine's base pressure is not good (~order of 10^{-3} Pa), and oxygen impurity in Ar gas may oxidize the W film. The W film may be contaminated by oxidation.

We add the following sentence.

(11 Page, 29 line) The resistivity of the W layer is much bigger than that of the bulk value ($\sim 5.6 \times 10^{-8}$), which may be caused by slight oxidization during sputtering.

REVIEWERS' COMMENTS:

Reviewer #2 (Remarks to the Author):

The authors have addressed all the previous comments and the manuscript is improved. This paper is now ready for publication.

Reviewer #3 (Remarks to the Author):

The authors have fully addressed my questions in the response letter. Therefore, I could recommend the publication of the manuscript in Communications Engineering.